# Immunomodulatory Effects of Fruiting Body Extract and Solid-State-Cultivated Mycelia of *Taiwanofungus camphoratus*

**DOI:** 10.3390/nu11092256

**Published:** 2019-09-19

**Authors:** Liang-Hung Lin, Ching-Hsin Chi, Xiao-Han Zhang, Ying-Ju Chen, Ming-Fu Wang

**Affiliations:** 1Department of Food and Nutrition, Providence University, Shalu Dist., Taichung 43301, Taiwan; linlianghung@gmail.com (L.-H.L.); gingerchi329@gmail.com (C.-H.C.); g1030021@gm.pu.edu.tw (X.-H.Z.); 2Division of Allergy, Immunology & Rheumatology, Taichung Tzu Chi Hospital, Buddhist Tzu Chi Medical Foundation, Tanzi Dist., Taichung 42743, Taiwan; 3Bachelor Program in Health Care and Social Work for Indigenous Students, College of Humanities & Social Sciences, Providence University, Shalu Dist., Taichung 43301, Taiwan

**Keywords:** *Taiwanofungus camphoratus*, macrophages, natural killer cell, IgG, IgE

## Abstract

*Taiwanofungus camphoratus* is a rare and valuable medicinal mushroom indigenous to Taiwan. It has traditionally been used to promote good health. This study aimed to explore the immunomodulatory effects of “Leader Deluxe *Taiwanofungus camphoratus* capsule” (LDAC). LDAC is a healthy food product composed of fruiting body extract and solid-state-cultivated mycelia of *T. camphoratus*. Two complementary studies were performed. In the first, LDAC was orally administered to BABL/c female mice for 6 weeks as part of a non-specific immune study. In the second, mice were treated with LDAC for 8 weeks and immunized with ovalbumin (OVA) in a specific immune study. LDAC increased the growth of splenic immune cells and enhanced the activity of macrophages and natural killer cells. It increased the levels of interleukin (IL)-2, interferon (IFN)-γ, serum immunoglobulin (Ig)G, and OVA-IgG, and decreased the levels of IL-4, IL-5, tumor necrosis factor (TNF)-α, serum IgE, and OVA-IgE. Thus, the findings of this study strongly supported the idea that LDAC possesses immunomodulatory activity.

## 1. Introduction

The immune system is important for protecting the human body from different types of pathogens and against tumor invasion. Maintenance of optimal immune system functions is crucial for good health. However, increasing stress and other factors, such as work pressure, pollution, aging, and irregular lifestyle, disturb immune functions [1]. Therefore, the consumption of foods with immunomodulatory effects might be an efficient approach to strengthen the immune system. Many medicinal mushrooms have been taken for health improvement or prevention of diseases for a long time, including *Taiwanofungus camphoratus*, *Ganoderma lucidum*, and *Agaricus blazei* Murrill, which are all known for their immunomodulatory properties [2]. By different extraction preparation of medicinal mushrooms, immunomodulatory effects are proven on certain immune cells.

*Taiwanofungus camphoratus* (syn. *Antrodia cinnamomea*, *Antrodia camphorata*) is a rare and valuable medicinal mushroom that only grows in the inner cavity of *Cinnamomum kanehirae* Hayata (Lauraceae), an evergreen tree species that is also indigenous to Taiwan. In the long history of traditional Chinese and folk medicine, *T. camphoratus* was well-known for treating severe liver disease. Taiwanese aborigines used it to protect from drug, food intoxication, and diarrhea [3,4,5]. *T. camphoratus* possesses different biological activities, including anti-cancer [6,7,8], hepatoprotective [9,10], anti-inflammatory [11,12], and immune-regulatory effects [13,14]. Many studies have shown that *T. camphoratus* can enhance the activity of dendritic cells, phagocytes, and natural killer (NK) cells [14,15]. Moreover, *T. camphoratus* has the capacity to regulate immune response through induction of the Th1 pathway and was shown to protect from asthma and avoid *Schistosoma mansoni* infection in mice [13,15,16]. Some bio-active contents of *T. camphoratus,* such as polysaccharides, triterpenoids, and nucleic acids [4,5], have been identified to contribute to this immunomodulatory activity. In a previous study, we have shown that the solid-state-cultivated mycelia of *T. camphoratus* can increase immune cell proliferation and enhance the activity of macrophages and NK cells in mice [17]. However, the fruiting body of *T. camphoratus* possesses more bioactive compounds than the mycelium. Therefore, the purpose of this study was to assess the immunomodulatory effects of non-specific and specific immune responses in mice that were feed by the combination of fruiting body extract and solid-state-cultivated mycelia of *T. camphoratus* (“Leader Deluxe *Taiwanofungus camphoratus* capsule”; LDAC).

In the present study, treatment of BALB/c mice with LDAC at doses of 172.2, 344.4, and 861.0 mg/kg stimulated the proliferation of immune cells, cytolytic activity of NK cells, and phagocytic activity of peritoneal macrophages, and increased the level of interleukin-2 (IL-2), interferon-γ (IFN-γ), and serum immunoglobulin (Ig)G. In addition, a decrease in the serum anti-ovalbumin (OVA) IgE level and an increase in the serum anti-OVA IgG2a level were observed. Collectively, these findings suggest that LDAC is a potential immunomodulatory agent for improving human health.

## 2. Materials and Methods

### 2.1. Test Sample Preparation

LDAC, the test sample is composed of cut-log-cultivated fruiting body extract and solid-state-cultivated mycelia of *Taiwanofungus camphoratus* (manufactured by the R&D Center of Taiwan Leader Biotech Corp., Taichung, Taiwan).

### 2.2. Animal Treatment

All female BABL/c mice were purchased from BioLASCO Taiwan Co., Ltd. (Taipei, Taiwan). The study protocol was carried out in strict accordance with the recommendations in the Guide for the Care and Use of Laboratory Animals of the National Institutes of Health and approved by the Animal Research Ethics Committee at Medgaea Life Sciences Ltd. (IACUC number MG101054 and MG101055).

The mice were maintained in the housing facility at 22 °C ± 3 °C and at a relative humidity of 60% ± 10% with a 12 h light/12 h dark cycle. The test mice were free to take food and water. For non-specific immune tests, 8-week-old mice were divided into 5 groups of 10 mice each. The vehicle control group was treated with sterile water. The test sample groups were feed sterile water-dissolved LDAC with three different concentrations (172.2, 344.4 and 861.0 mg/kg bw) by gavage every day for 6 weeks. The dose levels of LDAC were equivalent to 0.5-, 1-, and 2.5-fold human recommended dose (1680 mg/day). The positive control group was feed by a commercial health food with immunomodulatory effects (633.5 mg/kg bw).

For antigen-specific immune tests, 8-week-old mice were divided into 6 groups of 10 mice each. The 6 groups were the vehicle group treated with sterile water, 3 test groups treated with different concentrations (172.2, 344.4, and 861.0 mg/kg bw), positive control group, normal group. The mice were orally administered the test sample daily for 8 weeks. After being treated for 4 weeks, the mice were intraperitoneally injected with 6.25 µg OVA (Sigma-Aldrich, St. Louis, MO, USA) emulsified in complete Freund’s adjuvant (CFA). Two weeks later, previously OVA immunized mice were administered a second-time intraperitoneal injection of OVA emulsified with incomplete Freund’s adjuvant (IFA). The normal group was treated with sterile water but not OVA. During the studies, each mouse was subjected to daily monitoring of general health conditions and clinical signs of illness or discomfort. The mice were weighed once per week. Whole blood was sampled, and the spleen was harvested at the designed time for further analysis.

### 2.3. Cell Proliferation Assay

The spleens were removed from the mice and processed for further analysis. Approximately 1.0 × 10^7^ splenocytes were cultured in 96-well plates with RPMI 1640 medium and treated with 10 μg/mL lipopolysaccharide (LPS), 5 μg/mL concanavalin A (Con A), or 25 μg/mL OVA. The cells were incubated at 37 °C in a CO_2_ incubator for 72 h, and cell proliferation was measured at 490 nm (OD_490_) by using a CellTiter 96^®^ AQueous One Solution Cell Proliferation Assay Kit (Promega, Madison, WI, USA). The stimulation index (SI) was calculated using the formula: OD_490_ of mitogen-stimulated cells/OD_490_ of non-stimulated cells.

### 2.4. NK Cell Activity

Splenocytes (effector cells) were isolated from the mice and incubated with PKH67-labeled YAC-1 cells (target cells) at an effector: target (E/T) ratio of 5, 10, and 25, at 37 °C for 4 h. The cells were stained with propidium iodide (PI) for 10 min, and the proportion of dead YAC-1 cells was determined by flow cytometry. A higher ratio of dead YAC-1 cells represents more cytotoxic activity of NK cells.

### 2.5. Phagocytotic Activity

To determine the phagocytotic activity of macrophage in sample-treated mice, macrophages from the abdominal cavity were isolated and incubated with fluorescein-labeled opsonized *Escherichia coli* at a multiplicity of infection (MOI) of 12.5, 25, and 50, at 37 °C for 2 h. The phagocytic activity (%) of the peritoneal macrophages was measured by flow cytometry. The higher ratio of labeled macrophages which were detected by flow cytometry represents greater phagocytotic activity.

### 2.6. Cytokine Analysis

Approximately 5 × 10^5^ splenocytes were cultured in 24-well plates and treated with Con A, LPS, or OVA for 48 to 72 h. The supernatant was harvested and centrifuged at 250× *g* for 10 min at 4 °C. The concentration of IL-2, IL-4, tumor necrosis factor-α (TNF-α), IFN-γ, and IL-5 was measured using an ELISA cytokine assay kit (eBioscience, San Diego, CA, USA).

### 2.7. Antibody Production Test

Blood samples were collected from the mice at the end of the study. These samples were centrifuged at 300× *g* for 10 min, and the concentration of the antibodies (IgG, IgM, IgE, OVA-IgG1, OVA-IgG2a, and OVA-IgE) in the serum was detected by ELISA. The antibody levels were calculated as follows: ELISA units (EU) = (*A*_sample_ − *A*_blank_)/*A*_positive serum_ − *A*_blank_), where *A* is absorbance.

### 2.8. Statistical Analysis

Results are shown as mean ± standard deviation (SD). Statistical analysis was processed by one-way ANOVA, followed by Duncan’s test, using SPSS software 19.0 (New York, NY, USA). *p* < 0.05 was considered statistically significant.

## 3. Results

### 3.1. Effect of LDAC on the Proliferation of Spleen Immune Cells

The proliferative activity of spleen immune cells was induced by Con A, LPS, or OVA. In the non-specific immune study, the LDAC and positive control groups showed increased proliferation of immune cells that were stimulated by Con A and LPS in a dose-dependent manner. In the specific immune study, the LDAC and positive control groups showed increased proliferation of immune cells that were stimulated by Con A, LPS, and OVA, compared to the vehicle control group (Table 1).

### 3.2. Effect of LDAC on NK Cell Activity

To evaluate the stimulatory effect of LDAC on NK cells, the cytotoxic activity of NK cells against YAC-1 cells was measured. The proportion of dead YAC-1 cells at an E/T ratio of 5:1, 10:1, and 25:1 was significantly higher among all treatment groups, compared to the vehicle control (Figure 1). The results suggest that LDAC exerts a dose-dependent effect on enhancing NK cell activity.

### 3.3. Effect of LDAC on Phagocyte Activity

Peritoneal macrophages were isolated from LDAC-treated mice and incubated with fluorescein-labeled *E. coli* at an MOI of 12.5, 25, and 50, and the extent of phagocytosis was determined by flow cytometry. As shown in Figure 2, LDAC markedly increased the extent of phagocytosis in a dose-dependent manner, and a significant increase was observed in cells treated at 344.4 and 861 mg/kg, compared to the vehicle control.

### 3.4. Effect of LDAC on Cytokine Levels

The basal levels of cytokines showed no significant difference across all groups. In the non-specific immune study, IL-2 level was significantly elevated in response to Con A, compared to the vehicle control group. Elevated IFN-γ level was also observed after Con A or LPS treatment in the LDAC and positive control groups. Moreover, IL-4 and IL-5, and TNF-α showed a significant decrease in response to Con A and LPS, and Con A, respectively, in the LDAC group (Table 2). In the specific immune study, an increase in IL-2 in response to Con A and LPS and a decrease in IL-4, IL-5, and TNF-α in response to OVA were noted in the LDAC group (Table 3). There was a trend toward increased IFN-γ in all treatment groups.

### 3.5. Effect of LDAC on Serum Antibody Secretion

In the non-specific immune study, the level of serum IgG was significantly elevated in all treatment groups, compared to the vehicle control. A significant decrease in the serum IgE level was observed in all treatment groups. The serum IgM level showed no difference between the treatment and vehicle control groups (Table 4). In the specific immune study, OVA-IgG2a level showed a significant increase and OVA-IgE level showed a significant decrease in all treatment groups, compared to the vehicle control group (Figure 3).

## 4. Discussion

The present study showed that LDAC, which is composed of fruiting body extract and solid-state-cultivated mycelia of *T. camphoratus*, can regulate non-specific and specific immune responses in mice. No statistically significant differences were observed in body weight and spleen-to-body weight ratios between the LDAC and vehicle control groups (data not shown). No obvious clinical signs of illness or death were observed in mice during the study period. Moreover, previous studies have shown that LDAC exerted no genotoxicity and no reproductive and development toxicity up to a daily dose of 2800 mg/kg. A 90-day-long repeated dose study with rats also found no adverse effects; the no observable adverse effect dose level (NOAEL) was 2800 mg/kg [18]. The estimated daily intake (EDI) of LDAC is 1680 mg/day (equal to 28 mg/kg per day for a 60-kg adult), and the safety margin of LDAC is 100 (margin of safety = NOAEL/EDI = 2800/28). Therefore, LDAC is expected to be safe at a dose below the recommended daily intake.

*T. camphoratus* contains many bioactive compounds, although the specific chemical components of *T. camphoratus* vary based on the method of cultivation. The fruiting bodies of *T. camphoratus* contain abundant triterpenoids, especially ergostane-type, which exhibit potent pharmacological effects [9]. Our previous studies have demonstrated that the solid-state-cultivated mycelial powder of *T. camphoratus* can increase immune cell proliferation as well as the levels of serum OVA-IgG and OVA-IgM in response to Con A stimulation [17,19]. The present study also found that LDAC not only increases the proliferation of immune cells in response to Con A, LPS, and OVA, but also regulates the levels of serum IgG, OVA-IgG2a, IgE, and OVA-IgE. The immunomodulatory effective dose of LDAC, which contained a part of cut-log-cultivated fruiting body extract of *T. camphoratus*, was less than that of the solid-state-cultivated mycelia of *T. camphoratus*. Thus, different methods of cultivation can affect the efficacy of *T. camphoratus,* and the fruiting body extract exerts stronger immunomodulatory effects than the mycelia.

The elicited immune response includes cell-mediated and humoral immunity, which is predominantly regulated by Th1 and Th2 cells. Th1 cells primarily produce IFN-γ and IL-2, which work toward a cell-mediated response, whereas Th2 cells primarily produce IL-4, IL-5, and IL-10, which work toward humoral and allergic responses [20]. The present study found that LDAC increased the secretion of IL-2 and IFN-γ and decreased the levels of IL-4 and IL-5, suggesting that LDAC might induce Type 1 cytokines. Cheng et al. demonstrated that *T. camphoratus* modulated the expression of Type 1 cytokines in splenocytes, thereby inducing mice to develop the Th1 response [21]. Lu et al. showed that *T. camphoratus* extract induced dendritic cells and IFN-γ production, and promoted T cells to the Th1 pathway^15^. Moreover, polysaccharides in the water extract of *T. camphoratus* can enhance the anti-tumor efficacy of DNA vaccine in mice by facilitating specific Th1 response [22]. Based on these findings, we speculated that LDAC triggered Th1 cytokine secretion, promoting the Th1 cell response, and activated macrophages and NK cells. Another interesting observation is that LDAC decreased the serum IgE level while increasing IgG level. LDAC can not only act as an immune-stimulator to aid the host against infections but can also modulate allergic responses.

## 5. Conclusions

In conclusion, LDAC can regulate non-specific and specific immune responses, including increased immune cell growth, enhanced activities of phagocytosis and NK cell and serum antibody secretion. However, further studies are necessary to elucidate the underlying mechanisms. The present study provided detailed evidence to support the notion that LDAC exerts immunomodulatory effects.

## Figures and Tables

**Figure 1 nutrients-11-02256-f001:**
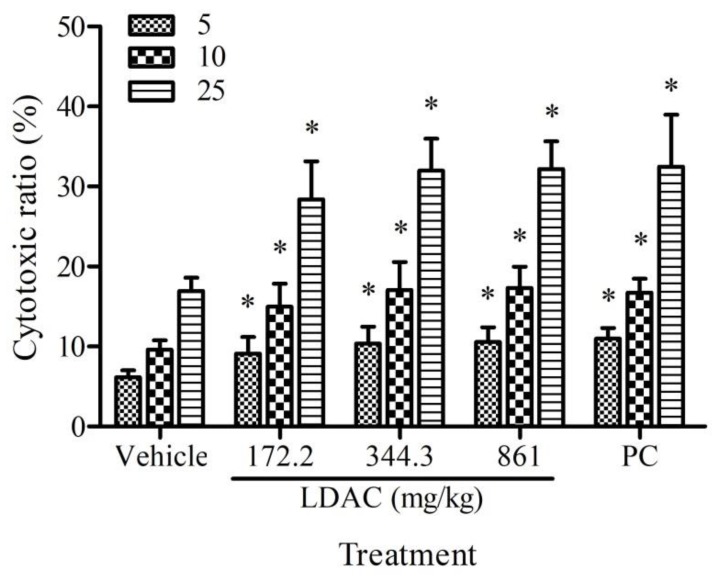
Effect of Leader Deluxe *Taiwanofungus camphoratus* capsule (LDAC) on cytotoxic activity of nature killer cells (NK). The mice were administered with LDAC (172.2, 344.4, 861 mg/kg) and positive control (PC, 633.5 mg/kg) for 6 weeks. The cytotoxic activity of natural killer cells was measured by incubation of effector cells (NK cells) with target cells (YAC-1 cells) at the ratio (E/T ratio) 5:1, 10:2, and 25:1. The percentage of dead YAC-1 cells represented the cytotoxic activity of NK cells. All data are presented as mean ± SD. * *p* < 0.05 compared to vehicle control.

**Figure 2 nutrients-11-02256-f002:**
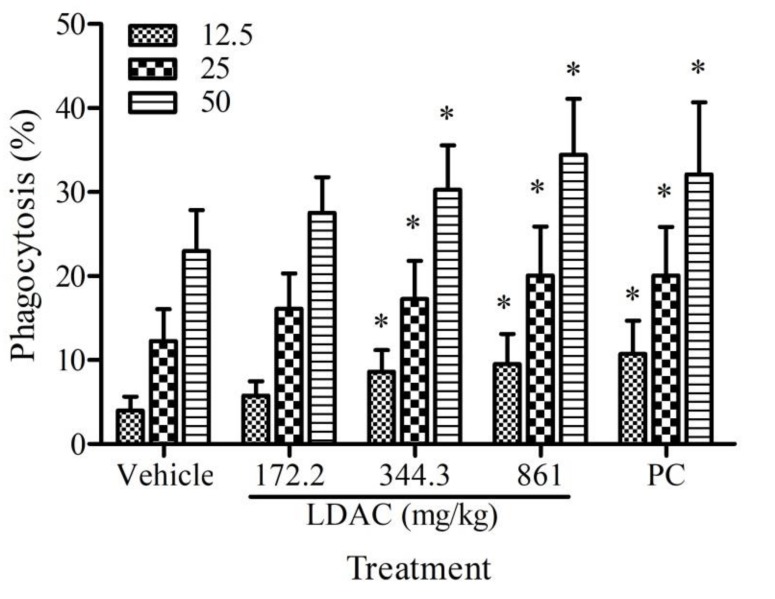
Effect of LDAC on phagocytic cell activity. The mice were administered with LDAC (172.2, 344.4, 861 mg/kg) and positive control (PC, 633.5 mg/kg) for 6 weeks. The peritoneal macrophages were isolated from mice and incubated with fluorescein-labeled *E. coli* at a multiplicity of infection (MOI) of 12.5–50. The ratio of phagocytosis was determined by flow cytometry. All data are presented as mean ± SD. * *p* < 0.05 compared to vehicle control.

**Figure 3 nutrients-11-02256-f003:**
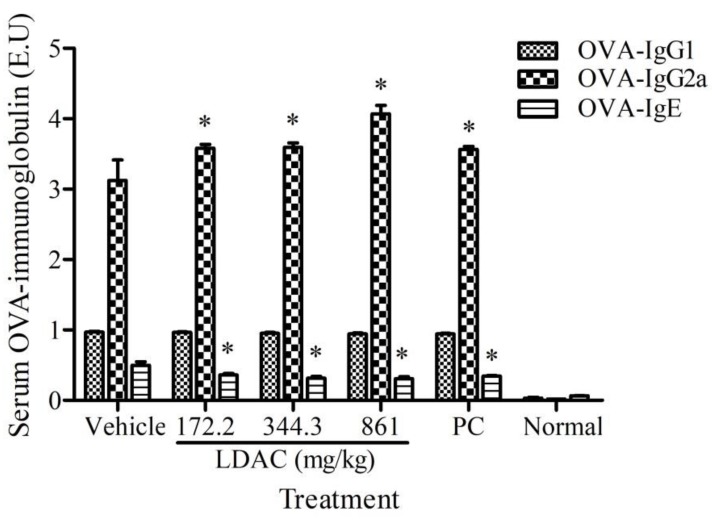
Effect of LDAC on serum ovalbumin (OVA)-IgG1, OVA-IgG2a, and OVA-IgE secretion. The OVA-immunized mice were administered with LDAC (172.2, 344.4, 861 mg/kg) and positive control (PC, 633.5 mg/kg) for 8 weeks. The serum OVA-IgG, OVA-IgG2a, and OVA-IgE were analyzed using ELISA assays. All data were presented as mean ± SD. * *p* < 0.05 compared to vehicle control. Normal group received only sterile water.

**Table 1 nutrients-11-02256-t001:** Proliferative capacity of spleen immune cells.

Group	Dose (mg/kg)	Stimulation Index (S.I.)		
		Con A 5 μg/mL	LPS 10 μg/mL	OVA 25 μg/mL
Non-specific immune test				
Vehicle	-	1.98 ± 0.19	2.15 ± 0.30	-
LDAC	172.2	2.22 ± 0.40	2.23 ± 0.31	-
LDAC	344.4	2.42 ± 0.72 *	2.54 ± 0.53	-
LDAC	861	2.95 ± 0.57 *	2.66 ± 0.54 *	-
PC	633.5	2.78 ± 0.40 *	2.86 ± 0.68 *	-
Specific immune test				
Vehicle	-	2.88 ± 0.25	2.47 ± 0.34	1.35 ± 0.18
LDAC	172.2	3.40 ± 0.35 *	2.72 ± 0.26	1.61 ± 0.28 *
LDAC	344.4	3.48 ± 0.18 *	2.78 ± 0.25 *	1.62 ± 0.31 *
LDAC	861	3.68 ± 0.44 *	2.92 ± 0.31 *	1.62 ± 0.21 *
PC	633.5	3.71 ± 0.21 *	3.30 ± 0.29 *	1.69 ± 0.37 *

All data are shown as mean ± SD. Vehicle: Vehicle control; PC: Positive Control; LDAC: Leader Deluxe *Taiwanofungus Camphoratus* capsule. Con A: concanavalin A; LPS: lipopolysaccharide; OVA: ovalbumin. * *p* < 0.05 compared to normal control.

**Table 2 nutrients-11-02256-t002:** Effect of Leader Deluxe *Taiwanofungus camphoratus* capsule (LDAC) on cytokines secretion in non-specific immune response.

Group	Dose (mg/kg)	Con A (7.5 μg/mL)	LPS (15 μg/mL)
		IL-2, pg/mL	
Vehicle		4545.2 ± 672.3	44.4 ± 7.4
LDAC	172.2	5227.6 ± 646.9 *	48.7 ± 5.3
LDAC	344.4	5257.9 ± 932.0 *	52.7 ± 7.9
LDAC	861	5333.2 ± 219.2 *	54.0 ± 8.6
PC	633.5	5152.9 ± 602.8	53.4 ± 17.0
		IL-4, pg/mL	
Vehicle		237.2 ± 63.9	10.4 ± 1.7
LDAC	172.2	189.1 ± 38.3 *	8.7 ± 1.1 *
LDAC	344.4	171.6 ± 45.2 *	8.1 ± 1.2 *
LDAC	861	149.6 ± 28.3 *	6.6 ± 1.3 *
PC	633.5	162.7 ± 29.1 *	8.3 ± 1.0 *
		IL-5, pg/mL	
Vehicle		197.7 ± 50.9	31.8 ± 6.3
LDAC	172.2	137.7 ± 20.0 *	25.1 ± 1.7 *
LDAC	344.4	111.1 ± 37.4 *	24.9 ± 2.1 *
LDAC	861	98.6 ± 9.0 *	23.4 ± 3.2 *
PC	633.5	101.4 ± 25.1 *	29.0 ± 3.1
		TNF-α, pg/mL	
Vehicle		208.9 ± 40.0	611.1 ± 183.1
LDAC	172.2	185.2 ± 21.1	594.7 ± 107.4
LDAC	344.4	171.5 ± 27.0 *	548.6 ± 95.2
LDAC	861	167.6 ± 29.5 *	523.4 ± 138.7
PC	633.5	193.6 ± 35.2	507.2 ± 95.1
		IFN-γ, pg/mL	
Vehicle		5512.6 ± 1920.7	2267.7 ± 807.1
LDAC	172.2	6128.6 ± 1744.5	3061.3 ± 962.6
LDAC	344.4	7912.1 ± 1745.1 *	4605.2 ± 1472.2 *
LDAC	861	8745.2 ± 2176.4 *	5038.4 ± 1874.5 *
PC	633.5	8439.9 ± 2441.8 *	4304.0 ± 1677.9 *

All data are presented as mean ± SD. Vehicle: Vehicle control; PC: Positive Control; LDAC: Leader Deluxe *Taiwanofungus Camphoratus* capsule; IL: interleukin; TNF-α: tumor necrosis factor-α; IFN-γ: interferon-γ. * *p* < 0.05 compared to vehicle control.

**Table 3 nutrients-11-02256-t003:** Effect of LDAC on cytokines secretion in specific immune response.

Group	Dose (mg/kg)	Con A (5 μg/mL)	LPS (10 μg/mL)	OVA (25 μg/mL)
		IL-2 (pg/mL)		
Vehicle	-	4808.3 ± 957.7	94.5 ± 26.6	205.1 ± 38.4
LDAC	172.2	5399.7 ± 704.7	107.5 ± 20.7	234.3 ± 31.6
LDAC	344.4	5464.6 ± 642.3	116.6 ± 31.5	217.2 ± 45.4
LDAC	861	5777.5 ± 1277.5 *	125.8 ± 31.5 *	205.2 ± 43.0
PC	633.5	5519.0 ± 632.5	87.8 ± 16.9	205.9 ± 50.6
Normal	-	3366.8 ± 373.3	37.3 ± 9.7	38.5 ± 16.5
		IL-4 (pg/mL)		
Vehicle	-	48.5 ± 22.4	4.3 ± 0.4	5.0 ± 1.1
LDAC	172.2	33.2 ± 18.1	3.5 ± 0.5 *	4.0 ± 0.9 *
LDAC	344.4	34.3 ± 16.1	3.3 ± 0.3 *	3.5 ± 0.5 *
LDAC	861	22.8 ± 11.3 *	3.5 ± 0.2 *	3.2 ± 0.3 *
PC	633.5	19.9 ± 7.6 *	3.4 ± 0.5 *	3.4 ± 0.7 *
Normal	-	44.5 ± 18.0	3.4 ± 0.3	2.6 ± 0.4
		IL-5 (pg/mL)		
Vehicle	-	65.0 ± 12.3	14.2 ± 4.7	8.4 ± 2.5
LDAC	172.2	57.0 ± 16.7	11.1 ± 6.5	7.7 ± 2.9
LDAC	344.4	63.2 ± 14.9	10.2 ± 3.3	6.6 ± 1.5 *
LDAC	861	58.9 ± 10.0	10.7 ± 2.8	6.5 ± 1.4 *
PC	633.5	53.8 ± 8.0	11.9 ± 5.7	5.9 ± 0.7 *
Normal	-	64.0 ± 21.3	13.5 ± 5.5	4.7 ± 0.4
		TNF-α (pg/mL)		
Vehicle	-	206.1 ± 84.5	583.9 ± 110.2	13.7 ± 6.8
LDAC	172.2	192.7 ± 71.0	430.2 ± 81.9 *	10.6 ± 5.8
LDAC	344.4	180.9 ± 58.3	396.5 ± 99.3 *	8.9 ± 4.7
LDAC	861	177.7 ± 78.4	394.4 ± 54.5 *	6.1 ± 4.7 *
PC	633.5	137.3 ± 73.0	419.6 ± 117.3 *	10.1 ± 6.1
Normal	-	140.6 ± 48.3	364.9 ± 76.1	4.9 ± 1.6
		IFN-γ (pg/mL)		
Vehicle	-	5128.0 ± 1927.1	1389.8 ± 736.4	1606.5 ± 758.9
LDAC	172.2	6649.7 ± 2202.2	1406.1 ± 199.5	1952.8 ± 958.3
LDAC	344.4	6970.9 ± 3132.8	1493.7 ± 255.2	1920.5 ± 661.3
LDAC	861	7139.3 ± 2550.0	1465.9 ± 192.4	1897.4 ± 751.3
PC	633.5	6525.7 ± 1991.0	1444.1 ± 626.0	1814.1 ± 360.9
Normal	-	4233.7 ± 1487.3	1274.0 ± 452.0	1050.6 ± 481.0

All data are presented as mean ± SD. Vehicle: Vehicle control; PC: Positive Control; Normal group received only sterile water but no OVA. LDAC: Leader Deluxe *Taiwanofungus Camphoratus* capsule. * *p* < 0.05 compared to vehicle control.

**Table 4 nutrients-11-02256-t004:** Serum antibody.

Group	Dose (mg/kg)	IgG (μg/mL)	IgM (μg/mL)	IgE (μg/mL)
Vehicle	-	511.8 ± 44.9	423.6 ± 80.4	0.29 ± 0.11
LDAC	172.2	629.1 ± 70.1 *	428.5 ± 62.4	0.17 ± 0.04 *
LDAC	344.4	687.6 ± 63.4 *	481.1 ± 149.8	0.14 ± 0.04 *
LDAC	861	669.7 ± 73.5 *	447.7 ± 79.0	0.15 ± 0.07 *
PC	633.5	665.0 ± 38.1 *	434.5 ± 91.4	0.13 ± 0.05 *

All data are presented as mean ± SD. Vehicle: Vehicle control; PC: Positive Control; LDAC: Leader Deluxe *Taiwanofungus Camphoratus* capsule; Ig: immunoglobulin. * *p* < 0.05 compared to normal control.

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
