# Peer review of "Immunomodulatory Effects of Fruiting Body Extract and Solid-State-Cultivated Mycelia of Taiwanofungus camphoratus"

_nutrients, 2019, doi:10.3390/nu11092256_

Round 1

Reviewer 1 Report

This manuscript by Lin and colleagues described the immunomodulatory effects of medicinal mushroom, T. camphoratus (LDAC) in a model of OVA-sensitized mice and ex vivo studies. The authors investigated several cellular assays including proliferation, phagocytic, cytokines and Igs assays. The authors presented a considerable amount of work however the conclusions from the existing data are far-reaching in terms for the beneficial effects of LDAC.

Major comments

Statement of ethic approval for animal study is missing Statement of competing interests or conflict of interest is needed, particularly LDAC is provided by a commercial entity There are several instances where the authors over-claimed their results/conclusions in this study There is no data reporting on NK activity in the manuscript (line 20, 53, 89, 124, etc), the authors only isolated splenocytes for their assay. To claim the cytotoxic activity of NK cells, a proper isolation of NK cells is needed. Please change the term ‘NK cells’ in the manuscript and figure legends. There is also no data reporting on immune cells proliferation (line 53 and 220), only splenocytes or splenic immune cells. In the discussion (line 19, 197), there is no evidence of LDAC increases proliferation of immune cells. The data presented in this manuscript are all ex vivo study of splenocytes. Please change accordingly. Line 181-183. The regulation of non-specific immune response of LDAC was only performed on ex vivo isolated splenocytes not in mice. There is no evidence of administration of LPS or Con A into mice in this study. Please change the statement. Line 199-203. Were the different methods of cultivation, mycelia or fruiting body extract compared in this study? Give evidence to support the statement or change accordingly Several techniques/assays described in the results were missing in the methods and materials Line 70. The positive control used in this study, ‘commercial health food with immunomodulatory effect’ is meaningless to readers or researchers. There is insufficient information to replicate or compare the study. Please state the positive control used. Please state the number of animals per groups in the antigen-specific study (line 73) Please state the methods for the isolation of splenocytes and macrophages Please state the source of CFA, IFA, Con A and labelled E. coli Please state the catalogue number, source and purity of LPS Please include a method section on flow cytometry and representative gating strategies for section 2.4 and 2.5

Minor comments

Cytolytic (line 53, 93, 125, 131, 134, etc) is the lysis of a cell due to osmotic imbalance. Cytotoxic would be a better word in this study Line 36, indigenous would be a better word for endemic Figure 1, how was the cytotoxic ratio % calculated? Please state in the methods Figure 2, how was phagocytosis % calculated? Please state in the methods What was the ‘normal group’ in table 3, 4 and figure 3? Please state in the methods

Author Response

Some words were unavoidable to repeat because of previous published articles of target sample, Taiwanofungus camphoratus, and sample was provided from commercial product.

Our article is to show the different outcomes in immunomodulatory activities.

Line 9: Add “college of Humanities & Social Sciences, Providence University” for correction

Line 11: Delete “Short title: Immunomodulation by Taiwanofungus camphoratus”

Line 14: Deleted: ,….

Line 16: Add: Two complementary studies were performed. In the first,

Line 16: Deleted: The

Line 17: Deleted: ,

Added: In the second,

Line 32:

Deleted: have long been consumed for their health benefits and disease-preventive properties,

Added: have been took for health improvement or prevention of diseases for a long time,

Line 34:

Added: By different extraction preparation of medicinal mushrooms, immunomodulatory effects are proven on certain immune cells.

Line 36-40: Re-write discovered function of test sample from published journals.

Line 45: Deleted: that can

Line 45: Deleted: inhibit

Added: and was shown to protect from asthma and avoid

Line 46: Deleted: Many bioactive components

Added: Some bio-active content

Line 51-54: Re-write for better reading.

Line 58: Deleted: Taken together

Line 67-70: Added: the code of study.

Line 72: Deleted: and free access to

Added: the test mice are free to take

Line 73:

Deleted: ; positive control group, with a commercial health food with immunomodulatory effects (633.5mg/kg bw); and test article groups

Line 74:

Deleted: The test article was dissolved in sterile water to prepare dosing solutions.

Added: The test sample groups were freed sterile water-dissolved LDAC with three different concentration (172.2, 344.4, and 861.0mg/kg bw) by gavage everyday for 6 weeks.

Line 81: Deleted: after the first OVA immunization, the

Line 84:

Deleted: Blood samples and spleen were collected

Added: Whole blood was sampled and spleen was harvested at the designed time

Line 97: Added was determined by flow cytometry.

Line 98:

Deleted: , which represents the cytolytic activity (%) of the NK cells, was determined by flow cytometry.

Line 115: Deleted: presented …. performed

Added: shown … processed

Line 123:

Deleted: a significantly higher ratio ….. was observed in the LDAC and positive control groups..

Added: the LDAC and positive control groups showed increased proliferation … that were

Line 126: Deleted: presented

Added: shown

Line 132: Deleted: exerted

Added: exerts

Line 159:

Deleted: Increased IFN-gamma level, although statistically insignificant, was also observed in all treatment groups.

Added: There was a trend toward increased IFN-gamma in all treatment groups.

Line 185: Deleted: composed

Added: which is composed

Line 207: Deleted: compared to

Line 296-299: Edit and delete some repeated words.

Line 313: re-edit words.

Line 316: red-edit words.

Line 378: Interferon-gamma was increased in all treatment groups, although not significant statistically but a trend was observed.

Line 407: Re-edit words.

Line 418-419: re-edit words.

Line 439: Re-edit for better reading.

Line 446: Re-edit for better reading.

Line 451: Re-edit for better reading.

Reviewer 2 Report

The authors Lin et al have made interesting observations to dissect the mechanism of action of the medicinal mushroom Taiwanofungus camphoratus. Although, authors have made commendable attempt at supporting their hypothesis that T camphoratus has immunomodulatory effects, however the following queries needs to addresses:

The authors should consider early time points for cytokine analysis, (6,12,18 hours) instead of 48-72 hours. It is not clear whether the effect of the extract is inclined towards lowering pro-inflammatory response OR leading to a faster recovery from the immune insult. The authors claim that the extract has immunomodulatory effects, however there is not sufficient data to support this hypothesis. Extensive assays include early and time points needs to be incorporated for their analyses. For proliferation of spleen cells, the authors should rather enrich a specfic subset of T cells (CD4 or CD8) and use them for further assays. There is no information for Gating-strategy employed for NK cells. Since these cell population do not have well-established markers, this piece of information is crucial to judge the viability of this assay. The authors should graph (radar plots) their data for better comparison and understanding of the different groups; and move the table with numbers to supplemental instead.

Author Response

Firstly, thanks for your kindly comments.

Point 1: The authors should consider early time points for cytokine analysis, (6,12,18 hours) instead of 48-72 hours

Response 1: Treat time 48-72 hours was followed procedure of eBioscience ELISA cytokine assay kit. (Line 115-118).

Point 2: The authors claim that the extract has immunomodulatory effects, however there is not sufficient data to support this hypothesis.

Response 2: In antigen-specific test, increased IL-2, decreased IL-4, IL-5 were observed in test-animal splenic cells after stimulation of Con A and LPS. (Line 174-176). These results were suggesting possible Th1 cytokine activity. Immunomodulatory effect was noted although further investigation should be done in the further tests.

Point 3: There is no information for Gating- strategy employed for NK cells. Since these cell population do not have well-established markers, this piece of information is crucial to judge the viability of this assay.

Response 3: Activity  of NK cell were detected by ratio of PKH67-labeled dead YAC-1 cells. (Line 103-107, Figure 1), although no detailed cell surface antigen maker data of T cell and NK cell.

Some words were unavoidable to repeat because of previous published articles of target sample, Taiwanofungus camphoratus, and sample was provided from commercial product.

Our article is to show the different outcomes in immunomodulatory activities.

Line 9: Add “college of Humanities & Social Sciences, Providence University” for correction

Line 11: Delete “Short title: Immunomodulation by Taiwanofungus camphoratus”

Line 14: Deleted: ,….

Line 16: Add: Two complementary studies were performed. In the first,

Line 16: Deleted: The

Line 17: Deleted: ,

             Added: In the second,

Line 32:

Deleted: have long been consumed for their health benefits and disease-preventive properties,

Added: have been took for health improvement or prevention of diseases for a long time,

Line 34:

Added: By different extraction preparation of medicinal mushrooms, immunomodulatory   effects are proven on certain immune cells.

Line 36-40: Re-write discovered function of test sample from published journals.

Line 45: Deleted: that can

Line 45: Deleted: inhibit

Added: and was shown to protect from asthma and avoid

Line 46: Deleted: Many bioactive components

Added: Some bio-active content

Line 51-54: Re-write for better reading.

Line 58: Deleted: Taken together

Line 67-70: Added: the code of study.

Line 72: Deleted: and free access to

Added: the test mice are free to take

Line 73:

Deleted: ; positive control group, with a commercial health food with immunomodulatory effects (633.5mg/kg bw); and test article groups

Line 74:

Deleted: The test article was dissolved in sterile water to prepare dosing solutions.

Added: The test sample groups were freed sterile water-dissolved LDAC with three different concentration (172.2, 344.4, and 861.0mg/kg bw) by gavage everyday for 6 weeks.

Line 81: Deleted: after the first OVA immunization, the

Line 84:

Deleted: Blood samples and spleen were collected

Added: Whole blood was sampled and spleen was harvested at the designed time

Line 97: Added was determined by flow cytometry.

Line 98:

Deleted: , which represents the cytolytic activity (%) of the NK cells, was determined by flow cytometry.

Line 115: Deleted: presented …. performed

Added: shown … processed

Line 123:

Deleted: a significantly higher ratio ….. was observed in the LDAC and positive control groups..

Added: the LDAC and positive control groups showed increased proliferation … that were

Line 126: Deleted: presented

Added: shown

Line 132: Deleted: exerted

Added: exerts

Line 159:

Deleted: Increased IFN-gamma level, although statistically insignificant, was also observed in all treatment groups.

Added: There was a trend toward increased IFN-gamma in all treatment groups.

Line 185: Deleted: composed

Added: which is composed

Line 207: Deleted: compared to

Line 296-299: Edit and delete some repeated words.

Line 313: re-edit words.

Line 316: red-edit words.

Line 378: Interferon-gamma was increased in all treatment groups, although not significant statistically but a trend was observed.

Line 407: Re-edit words.

Line 418-419: re-edit words.

Line 439: Re-edit for better reading.

Line 446: Re-edit for better reading.

Line 451: Re-edit for better reading.

Round 2

Reviewer 1 Report

The authors made a very poor attempt to address this reviewer comments and concerns about the study. Most of the comments were ignored and the authors only made minor editorial changes to the manuscript. I have copied my comment from the first review to re-emphasize the comments.

Major comments

Statement of competing interests or conflict of interest is needed, regardless of LDAC is purchased or provided by a commercial entity There are several instances where the authors over-claimed their results/conclusions in this study There is no data reporting on NK activity in the manuscript, the authors only isolated splenocytes for their assay. To claim the cytotoxic activity of NK cells, a proper isolation of NK cells is needed. Please change the term ‘NK cells’ in the manuscript and figure legends. There is also no data reporting on immune cells proliferation (line 67 and 247), only splenocytes or splenic immune cells. In the discussion (line 224), there is no evidence of LDAC increases proliferation of immune cells. The data presented in this manuscript are all ex vivo study of splenocytes. Please change accordingly. Line 208. The regulation of non-specific immune response of LDAC was only performed on ex vivo isolated splenocytes not in mice. There is no evidence of administration of LPS or Con A into mice in this study. Please change the statement. Several techniques/assays described in the results were missing in the methods and materials Line 94. The positive control used in this study, ‘commercial health food with immunomodulatory effect’ is meaningless to readers or researchers. There is insufficient information to replicate or compare the study. Please state the positive control used. Please state the methods for the isolation of splenocytes and macrophages Please state the source of CFA, IFA, Con A and labelled E. coli Please state the catalogue number, source and purity of LPS Please include a method section on flow cytometry and representative gating strategies for section 2.4 and 2.5 Figure 1, how was the cytotoxic ratio % calculated? Please state in the methods Figure 2, how was phagocytosis % calculated? Please state in the methods

Author Response

Dear Prof. Reviewer 1:

Firstly, thanks for your kindly comments.

For Major Comments:

Point 1: Statement of ethic approval for animal study is missing Statement of competing interests or conflict of interest is needed, particularly LDAC is provided by a commercial entity There are several instances where the authors over-claimed their results/conclusions in this study

Response 1: Added statement of ethic approval of animal study was added in (Line 71-73), including approval document with reference number by Animal Research Ethics Committee.

Point 2: There is no data reporting on NK activity in the manuscript (line 20, 53, 89, 124, etc), the authors only isolated splenocytes for their assay. To claim the cytotoxic activity of NK cells, a proper isolation of NK cells is needed. Please change the term ‘NK cells’ in the manuscript and figure legends. There is also no data reporting on immune cells proliferation (line 53 and 220), only splenocytes or splenic immune cells

Response 2: There are several methods with commercial kits to detect cell-mediated cytotoxic activity of natural killer cells. In our study, we use YAC-1 assay. Activity expression of NK cells is described by detection of ratio of dead YAC-1 cells. (Line 97). This method is modified from the article : Thomas, L.M; Perteson M.E.; Long E.O.. Cutting Edge: NK Cell Licensing Modulates Adhesion to Target Cells. J Immunol. 2013, 191 (8) 3981-3985; DOI: https://doi.org/10.4049/jimmunol.1301159

Point 3: The regulation of non-specific immune response of LDAC was only performed on ex vivo isolated splenocytes not in mice. There is no evidence of administration of LPS or Con A into mice in this study. Please change the statement. Line 199-203.

Response 3: Although stimulation of Con A and LPS were not added into mice, the harvested splenic cells treated ex vivo actually showed statistically significant difference of cell proliferation in high dose group and positive control group, compared to negative control group. (Line 123-124, and Table 1).  

Point 4: Give evidence to support the statement or change accordingly Several techniques/assays described in the results were missing in the methods and materials Line 70.

Response 4: In Table 2, immunological cytokines (IL-2, IL-4, IL-5, TNF-α) were detected by ELISA cytokine assay kit (Line 105-108). The results were described in Line 155-160.  

Point 5: Please state the number of animals per groups in the antigen-specific study (line 73)

Response 5. Number of animals in antigen-specific group were added in Line 82-85.

For Minor Comments:

Point 1: Cytolytic (line 53, 93, 125, 131, 134, etc) is the lysis of a cell due to osmotic imbalance. Cytotoxic would be a better word in this study Line 36.

Response 1: Correct “cytolytic” to “cytotoxic” in Line 131, Line (138, 141 in Figure 1). Cytotoxic activity of NK cell is better description of relationship to death of  YAC-1 cell after treatment.

Point 2: indigenous would be a better word for endemic.

Response2 : Correct “endemic” to “indigenous” (Line 12, 38).

Point 3: Figure 1, how was the cytotoxic ratio % calculated?

Response 3: In Figure 1, Cytotoxic ratio, means percentage of dead YAC-1 cells, which is labeled by PKH67 treatment and is detected by flow cytometry (Line 99-100, Materials and Methods, 2.4 NK cell activity) .

Point 4: Please state in the methods Figure 2, how was phagocytosis % calculated? Please state in the methods

Response 4: In Figure 2, numbers of fluorescein-labeled macrophage was detected by flow cytometry and data showed with ratio of detected cells. (Line 105-107, Materials and Methods, 2.5 Phagocytotic activity).

Point 5: Please state in the methods What was the ‘normal group’ in table 3, 4 and figure 3? Please state in the methods

Response 5: “Normal group” in Table 3, Table4 and Figure 3: Normal group means, in antigen-specific immune test, the 6th group mice was treated with sterile water but not OVA like other five test groups. (Added in Line 85, Materials and Methods, 2.2 Animal treatment).
